# Tumor Infiltrating Lymphocytes across Breast Cancer Subtypes: Current Issues for Biomarker Assessment

**DOI:** 10.3390/cancers15030767

**Published:** 2023-01-26

**Authors:** Carmine Valenza, Beatrice Taurelli Salimbeni, Celeste Santoro, Dario Trapani, Gabriele Antonarelli, Giuseppe Curigliano

**Affiliations:** 1Division of New Drugs and Early Drug Development for Innovative Therapies, European Institute of Oncology, IRCCS, Via Ripamonti 435, 20141 Milan, Italy; 2Department of Oncology and Hematology-Oncology, University of Milan, Via Festa del Perdono 7, 20122 Milan, Italy; 3Department of Clinical and Molecular Medicine, Sapienza University of Rome, Piazzale Aldo Moro 5, 00185 Rome, Italy

**Keywords:** breast cancer, triple negative breast cancer, breast cancer subtypes, tumor infiltrating lymphocytes, immune checkpoint inhibitors, biomarker, tumor microenvironment

## Abstract

**Simple Summary:**

Tumor-infiltrating lymphocytes (TILs) are immune cells that can be involved in the anti-tumor response and are presently viewed as a promising, inexpensive biomarker with prognostic and predictive potential. It has been demonstrated that patients with early triple-negative breast cancer and high TILs report higher responses to treatments with improved survival outcomes. Moreover, TIL expression may have a prognostic role in non-triple-negative subtypes and predict the response to immuno-oncology agents. However, the use of TILs as biomarkers in clinical practice is still limited by the lack of rigorous, prospective validation. This review summarizes the most important current issues and future challenges to assessing and validating TILs as predictive and prognostic biomarkers in patients with breast cancer.

**Abstract:**

Tumor-infiltrating lymphocytes (TILs) represent a surrogate biomarker of anti-tumor, lymphocyte-mediated immunity. In early, triple-negative breast cancer, TILs have level 1B of evidence to predict clinical outcomes. TILs represent a promising biomarker to select patients who can experience a better prognosis with de-intensified cancer treatments and derive larger benefits from immune checkpoint inhibitors. However, the assessment and the validation of TILs as a biomarker require a prospective and rigorous demonstration of its clinical validity and utility, provided reproducible analytical performance. With pending data about the prospective validation of TILs’ clinical validity to modulate treatments in early breast cancer, this review summarizes the most important current issues and future challenges related to the implementation of TILs assessments across all breast cancer subtypes and their potential integration into clinical practice.

## 1. Introduction

Breast cancer (BC) is the most common cancer in women and represents the leading cause of death from all cancers [1,2]. BC is traditionally classified according to the expression of hormonal receptors (HR) and human epidermal growth factor receptor 2 (HER2) in three subtypes: HR-positive (HR+)/HER2-negative (HER2-) BC, HER2+ BC, and HR-/HER2- or triple-negative breast cancer (TNBC), which account for approximately 70%, 15–20%, and 10–15% of all BC diagnoses, respectively [3].

Immunotherapy (IO) drugs, and in particular immune-checkpoint inhibitors (ICIs), have revolutionized the treatment landscape of several cancer types, including TNBC, despite BC having been historically considered immunogenically quiescent and hence less likely to derive benefit from IO approaches [4]. In particular, ICIs have demonstrated activity in patients with TNBC in the metastatic setting if administered in combination with first-line chemotherapy in patients with programmed death ligand 1 (PD-L1) positive cancers and in the early setting in combination with neoadjuvant chemotherapy, regardless of PD-L1 status [5].

Despite widespread adoption and current clinical use across a wide range of diseases, PD-L1 is still an imperfect immune biomarker that does not accurately depict the immune landscape of BC or allow for the best patient selection and stratification, particularly in non-metastatic settings [5].

In fact, the improvement of outcomes through the optimization of therapeutic strategy requires validated biomarkers for patient selection and to guide treatment (de)escalation. The introduction of a biomarker requires the demonstration and assessment of: (i) the assays’ technical performance, namely analytical validity; (ii) the ability of the biomarker to identify relevant clinical subgroups with specific prognostic and/or predictive features, namely clinical validity; and (iii) the ability of the biomarker to guide treatment decisions by providing a clinical benefit, namely clinical utility [6].

Tumor-infiltrating lymphocytes (TILs) represent a surrogate biomarker of anti-tumor, lymphocyte-mediated immunity, whose activation is related to better outcomes in patients with TNBC [5]. Enrichment in TILs may identify highly immunogenic and immune-vulnerable BCs, as well as patients with a better prognosis who may need de-intensified treatments, for example, less chemotherapy or no chemotherapy. The clinical validity of TILs, in particular as prognostic biomarkers, has been widely assessed in the triple-negative subtype [5].

This narrative review aims at summarizing the prognostic and predictive role of TILs in patients with BC across all the clinical subtypes and highlighting the most important challenges for the assessment of this biomarker, including as it pertains to issues of analytical validity, clinical validity, and clinical utility of TILs.

## 2. TILs in BC: Biological Significance, Assessment and Analytical Validity

TILs are defined as the sum of all mononuclear cells, including lymphocytes and plasma cells, in the tumor identified by classic morphological criteria; their presence in the tumor is thought to reflect the ongoing anti-tumor host immune response [7]. In fact, according to a fascinating but non-reproducible classification, a tumor can be considered "hot" or "inflamed" if characterized by an abundant immune-infiltrate, "intermediate" or "immune-excluded" if lymphocytes are present in the peritumoral stroma only, or "cold" or "immune-desert" if it is largely devoid of lymphocytes [8,9]. The determinants of these different tumor microenvironments (TMEs) have to be clarified, but it has been demonstrated that neoantigen release, tumor mutational burden (TMB), and tumor clonal heterogeneity are related to the intensity of the anti-tumor host immune response [7].

TILs can be further characterized according to their distribution in the TME: TILs are localized both in the stromal (stromal TILs [sTILs]) and in the intra-epithelial compartment (TILs with direct cell-to-cell contact with carcinoma cells). Furthermore, TILs can be classified according to their phenotype and biological function as CD8+ T cells, CD8+ tissue-resident memory T (TRM) cells, CD4+ T helper 1/2/17 (Th1, Th2, and Th17) cells, CD4+ regulatory T cells (Treg, defined by the expression of Forkhead Box P3 protein [FOXP3]), CD4+ follicular helper (Tfh) T cells, and tumor-infiltrating B cells [10]. Despite this functional and morphological heterogeneity of TILs, the identification of single immune subpopulations still retains a limited clinical value and should be further analyzed and correlated with different BC subtypes [11].

In order to standardize TIL assessment and improve the consistency and reproducibility for future studies, the *International TILs Working Group* developed a standardized methodology for the visual assessment of TILs in cancer specimens in 2014 [11]. According to the working group recommendations, TILs should be assessed in haematoxylin and eosin (H&E)-stained tumor sections and quantified as a ratio between the stromal area occupied by lymphocytes and plasma cells and the total intratumoral stromal area. Therefore, TILs should be assessed as a continuous parameter (i.e., % sTILs), because no clinically relevant TIL threshold has been identified yet; nonetheless, the term lymphocyte-predominant BC (LPBC) is usually used to describe BC with more than 50% of stromal lymphocytic infiltration [11].

Intra-epithelial TILs (iTILs) are not included in this practical definition because they are more difficult to evaluate and do not provide additional predictive or prognostic information compared to sTILs; indeed, since iTILs tend to be lower than sTILs, it has been hypothesized that they may represent an inactive, tissue-resident T-cell sub-population [11].

The inter-pathologist concordance of the visual and visual method proposed by the *International TILs Working Group* is very high (up to 86%), suggesting that it is an analytically reproducible biomarker [12]. Nevertheless, some concerns still need to be clarified: for example, the definition of the stromal area as the sole stroma abutting the tumor or as the stroma within the total tumor area; the evaluation of tumors with minimally assessable stroma; the risk of inclusion of other inflammatory cells (for instance, the unequivocal differentiation of a myofibroblast or an invasive lobular breast cancer cell with a lymphocyte is not always possible on an H&E slide) [7,13].

New molecular and computational techniques are under evaluation to better assess and dissect TILs in BC; however, nowadays, the H&E-based morphological method is the only standardized approach. Of note, the analytical validity of this approach cannot be formally demonstrated because of the lack of a gold standard for comparison [7].

## 3. TILs in BC: Clinical Validity across BC Subtypes

### 3.1. TILs in Triple Negative Breast Cancer

TNBC is considered the most immunogenic subtype of BC due to higher levels of TILs commonly reported, higher TMB, and enhanced PD-L1 expression compared to the other subtypes [5,14].

#### 3.1.1. The Prognostic Role of TILs in Early TNBC

Two large pooled analyses investigated the prognostic role of TILs in patients with early TNBC and provided level 1B evidence for the clinical validity of TILs; in fact, in both studies, TILs were retrospectively analyzed on archived tissues from independent prospective cohorts of randomized clinical trials [7,11].

The first pooled analysis, conducted by the German Breast Cancer Group, included 906 women with primary TNBC treated with neoadjuvant chemotherapy (NACT) in six randomized trials. TILs were analyzed both as a continuous parameter and in three predefined groups: low (sTILs: 0–10%), intermediate (sTILs: 11–59%), and high (sTILs: ≥60%). TILs have been demonstrated to predict responses to NACT and clinical benefit in terms of efficacy. In fact, pathological complete response (pCR) was achieved in 80 (31% of 260 patients) with low TILs, 117 (31% of 373) with intermediate TILs, and 136 (50% of 273) with high TILs (*p* < 0.0001, χ^2^ test for trend). In the univariable analysis, a 10% increase in TILs was associated with longer disease-free survival (DFS) (hazard ratio [HR]: 0.93; 95% CI: 0.87–0.98; *p* = 0.011) and overall survival (OS) (HR: 0.92; 95% CI: 0.86–0.99; *p* = 0.032) [15].

The second pooled analysis, which included 2148 patients from nine trials of adjuvant chemotherapy, also confirmed the role of sTILs in predicting invasive DFS (iDFS), distant DFS (DDFS), and OS: each 10% increment in sTILs corresponded to an HR of 0.87 (95% CI: 0.83–0.91) for iDFS, 0.83 (95% CI: 0.79–0.88) for DDFS, and 0.84 (95% CI: 0.79–0.89) for OS. Furthermore, in patients with node-negative disease and sTILs ≥ 30%, 3-year OS was 99%; while TILs were significantly reduced in patients with older age, larger tumor size, increased nodal involvement and lower histologic grade [16].

Concerning the prognostic value of TILs in untreated patients with early TNBC, Park et al. retrospectively demonstrated in four multicentre cohorts, including 476 patients, that both the presence of sTILs at baseline and a 10% increase in sTILs were independent prognostic factors of OS, iDFS, and DDFS. Considering patient with stage I TNBC and TILs ≥ 30%, 5-years OS was of 98% and 5-years DDFS of 97% [17]. Similarly, De Jong et al. retrospectively analyzed sTILs in 441 untreated young (<40 years) patients with pN0 TNBC (90% of them with pT1c or pT2 tumors) from the prospective Netherlands Cancer Registry, distinguishing patients with high sTILs (≥75%) from those with low sTILs (<30%). The authors showed an excellent 15-year cumulative incidence of a distant metastasis or death in patients with high sTILs (2.1%; 95% CI: 0–5.0), as compared to patients with low sTILs (38.4%; 95% CI: 32.1–44.6). Moreover, every 10% increment of sTILs was accompanied by a decreased risk of death of 19% (adjusted HR: 0.81; 95% CI: 0.76–0.87) [18].

Based on these studies, TILs in early TNBC have level 1B evidence as a prognostic biomarker. Accordingly, the World Health Organization (WHO) classification of tumors (5th edition) supports TIL quantification in TNBC and HER2+ subtypes. However, the *St. Gallen International Breast Cancer Consensus* from 2021 and the current clinical guidelines for breast cancer treatment do not endorse TILs as routine pathological markers and do not recommend de-escalation of chemotherapy based on TILs [7,19].

#### 3.1.2. TILs to Predict Treatment Effect of Immunotherapy

As far as the role of TILs in predicting activity and efficacy of IO agents is concerned, many data are emerging from both advanced and early settings, which differ for TME and sTILs infiltration. Indeed, the TME of advanced TNBC shows more depletion of immune effector cells with lower sTILs as compared to matched primary tissue [20].

The immune biomarker analysis of the IMpassion 130 phase 3 clinical trial, which randomized patients with metastatic TNBC to receive first line nab-paclitaxel with atezolizumab or placebo, showed a benefit among the two arms in terms of progression-free survival (PFS) and OS in PD-L1+/CD8+ patients (vs. PD-L1+/CD8-), with no differences in PFS and OS between CD8+ and CD8- patients regardless of PD-L1 status [20].

sTILs levels were associated with PD-L1 status but did not appear to impact the PFS or OS in patients from the control arm, while the efficacy of atezolizumab in terms of PFS and OS was predicted in patients from the experimental arm with PD-L1+ mTNBC. sTILs have been associated with higher response rates and improved clinical outcomes with ICIs in mTNBC, as emerged in immune biomarkers analyses from KEYNOTE-086 and KEYNOTE-119 trials with pembrolizumab as a single agent in metastatic TNBC [21,22,23].

Concerning early TNBC, STIL’s role in predicting responses to IO agents has not been fully defined. For example, in the phase 2 GeparNuevo clinical trial, which randomized patients to receive neoadjuvant chemotherapy plus durvalumab or a placebo, sTILs predicted higher pCR rates in both arms [24,25]. A multivariate analysis showed that the increase in iTILs from baseline to post-window phase samples is independently associated with high pCR rates in the IO group (OR: 9.36; 95% CI: 1.26–69.65; *p* = 0.029), but not in the placebo group (OR: 1.22; 95% CI: 0.65–2.27; *p* = 0.540).

Similarly, both in the phase 1b KEYNOTE-173 and the phase 2 randomized I-SPY2 clinical trials, which respectively evaluated the combination of pembrolizumab plus neoadjuvant chemotherapy in early TNBC and HER2-negative BC, a correlation between baseline sTILs or T cell density and pCR emerged [26,27]. Moreover, some intriguing data come from the dynamic evaluation of TILs: for example, in NeoTRIPaPDL1, sTILs assessed at day 1 of cycle 2 were suggested to be an early surrogate of pCR (sTILs ≥40% had 71.4% pCR, OR: 6.38, 95% CI: 2.24–20.9, *p* = 0.0007) [28].

These data may suggest a predictive value of TILs in the response to ICI, but their interpretation requires caution. In fact, most of these immune biomarker analyses were exploratory, and TILs were not included among stratification factors, apart from the GeparNuevo trial, in which sTILs were not specifically related to durvalumab response. Furthermore, the prognostic and predictive role of TILs requires independent prospective validation in randomized clinical trials.

### 3.2. TILs in HER2+ Breast Cancer

Many studies evaluated the role of TILs in early HER2+ BC and demonstrated that increased TILs positively correlate with response to neoadjuvant therapy and with clinical benefit in HER2-positive BC, similarly to what was observed in TNBC [15,29,30].

Heppner et al. assessed TILs in HER2+ samples from the GeparQuattro (NCT00288002) and GeparQuinto (NCT00567554) trials. They found out that patients with HER2+, LPBC (sTILs >60%) had a higher rate of pCR compared to non-LPBC-type [31]. A secondary analysis of patients with early HER2+ BC treated with neoadjuvant trastuzumab and lapatinib in the NeoALTTO trial showed that baseline TILs represent an independent positive prognostic factor for both pCR and event-free survival (EFS). Intriguingly, patients with a high TILs percentage at baseline have very favorable EFS regardless of pCR outcome [32]. Similarly, in the PAMELA trial, which assessed the response of neoadjuvant lapatinib and trastuzumab in patients with HER2+ BC, TILs at day 15 of treatment were shown to be an independent predictive marker of pCR [33].

The prognostic and predictive roles of TILs were also assessed in the adjuvant setting. Indeed, distant disease-free survival (DDFS) was longer in patients with high TILs levels both in the FinHER trial and the ShortHER trial [34,35]. In the former, HER2-positive BC patients were randomized to receive adjuvant chemotherapy with or without trastuzumab, and each 10% TIL increase was significantly associated with decreased distant recurrence in patients in the trastuzumab arm. Instead, the ShortHER trial, which failed to demonstrate the non-inferiority of 9 weeks of adjuvant trastuzumab vs. the standard 12 months of treatment, showed that an increase in the TILs was independently associated with better DDFS in a multivariate model (HR: 0.73, 95% CI: 0.59–0.89, *p* = 0.006, for each 10% TILs increment).

Nonetheless, in contrast to these data, in the analysis from the N9831 trial, higher TILs correlated with better recurrence-free survival (RFS) in the chemotherapy arm but not in the chemotherapy plus trastuzumab arm, where, unexpectedly, they predicted a lack of response to the anti-HER2 drug [36]. However, an immune gene expression analysis on the same samples showed a strong association with benefit [37].

Anyway, two large meta-analyses and a pooled analysis confirmed that high TILs at baseline were predictive of response to neoadjuvant chemotherapy plus trastuzumab in terms of pCR. One meta-analysis confirmed the association regardless of the type of neoadjuvant regimen [16,38,39]. As far as TILs’ role in predicting efficacy outcomes is concerned, a statistically significant correlation between high TILs at baseline and EFS emerged (pooled analysis by Denkert et al.: HR: 0.94; 95% CI: 0.89–0.99; *p* = 0·017. Gao et al. meta-analysis: HR: 0.94; 95%CI: 0.90–0.98; *p* = 0.003), while only the meta-analysis by Gao et al. reported a significant correlation with OS (HR: 0.91; 95% CI: 0.87–0.96; *p* < 0.001).

In conclusion, high stromal TILs correlate with an improved prognosis but also an enhanced response to trastuzumab and chemotherapy [34,40,41,42].

TILs’ prognostic and predictive roles in advanced disease are less well established. Luen et al. performed a retrospective analysis of the patients enrolled in the CLEOPATRA trial. They evaluated sTILs in the pre-treatment samples, observing that higher sTILs values were significantly associated with increased OS, irrespective of the treatment group. One explanation for the findings is that a more functional antitumor T-effector cell response may drive some of the effects of HER2-directed therapies, which appear to have anticancer activity by inhibiting HER2-signaling and recruiting immune-competent cells [43]. Indeed, 3-year OS estimates were 50% (95%CI: 44–57%) in patients with low TILs (≤20%) versus 55% (95% CI: 46–65%) in patients with high TILs (>20%) in the placebo group, and 64% (95%CI: 58–70%) versus 78% (95%CI: 69–87%) in patients with high and low TILs, respectively, in the pertuzumab group. Of note, only 7% of the samples analyzed were from metastatic sites, and the high threshold of 20% for sTILs was used because it was the closest decile to the mean sTIL value in the study population. Otherwise, a specific cut-off to define the rate of high or low TILS has not been defined in advanced BC disease so far.

Conversely, in the secondary analysis of the MA.31 phase 3 trial, no significant prognostic or predictive effects were found for TIL counts. In this trial, HER2+ BC patients were randomized to receive trastuzumab or lapatinib in association with taxane-based chemotherapy in the first-line setting. Noteworthy, a lower threshold for the identification of high TILs (5%) has been utilized in this study [44].

These data about the immunogenicity of HER2+ BC paved the way for clinical trials evaluating ICIs in this subtype, with disappointing results thus far [45,46,47]. For example, the PANACEA trial evaluated the combination of pembrolizumab and trastuzumab in patients with HER2+ metastatic BC progressing on trastuzumab. The objective response rate (ORR) was 15.2% and 0% in PD-L1+ and PD-L1- patients, respectively. TILs were higher in PD-L1+ patients and positively correlated with ORR. These data suggest that patients with elevated expression of TILs or PD-L1 could most likely benefit from combination therapy with ICIs and trastuzumab-based treatments [46].

However, in the KATE2 trial, also the combination of atezolizumab and T-DM1 failed in improving PFS and OS in unselected patients. In fact, the median PFS was 8.2 vs. 6.8 months (HR: 0.82; 95%CI: 0.55–1.23; *p* = 0.33) in the intention-to-treat population (ITT), while it was 8.5 vs. 4.1 months in PD-L1+ patients (HR 0.60; 95%CI: 0.32–1.11; *p* = 0.099). Of note, patients with high TILs (≥5%) had longer PFS in the atezolizumab group, while in the placebo group, it was the opposite [47]. Indeed, results from the KATE3 trial, which randomized only patients with PD-L1+ HER2+ breast cancer to T-DM1 with or without atezolizumab (NCT04740918), are awaited.

These controversial data from the metastatic setting need further evaluation and are burdened by several methodological concerns. In fact, these retrospective analyses differ in the TILs thresholds utilized, sampling (primary vs. metastatic site), and patients’ characteristics, including previous lines of therapy, baseline PD-L1 expression, and TILs value. Hence, the prospective assessment of the prognostic and predictive role of TILs in patients with metastatic HER2+ BC, as well as the definition of a threshold for high TILs, requires randomized phase 3 clinical trials with TILs as a stratification factor.

Lastly, as far as the role of TILs in predicting responses to IO agents is concerned, clinical trials showed poor results with immunotherapy in advanced disease, even when TILs enriched cancers [45,46,47], highlighting the need to further explore the complexity of interactions between different components of the immune system in HER2+ TME in order to further enrich potential responders.

### 3.3. TILs in HR+/HER2- Breast Cancer

HR+/HER2- BC is considered the least immunogenic subtype of BC; in fact, it is associated with a lower mean TILs count and a lower TMB. This may be due to estrogen receptor (ER) expression, which has been shown to correlate with higher Th2 infiltration, decreased MHC class II expression on BC cells, suppression of interferon-γ (IFN-γ) signaling and decreased cytolytic function of CD8+T-cells [7,14,48].

Nevertheless, the heterogeneity of immune TME in HR+/HER2- BC is high, and even the identification of a tiny "immunogenic" subgroup may have a significant impact in clinical practice because of the highest prevalence of this subtype of disease.

As far as the prognostic role of TILs in early HR+/HER2- BC is concerned, evidence from large retrospective studies is discordant: while some studies did not show a prognostic role of TILs in this subtype [49,50], others displayed an association between high TILs and a worse prognosis in terms of risk of recurrence and OS [51,52]. This discordance could be attributed to a number of methodological issues, such as the heterogeneity of the tested samples and the TIL assessment and quantification statistical methods.

A pooled analysis including 832 patients and a meta-analysis of 4 studies with 2836 patients assessed the prognostic value of TILs in patients with HR+/HER2- early BC who received neoadjuvant chemotherapy. In the first one, conducted by the German Breast Cancer Group, the increase in TILs was associated with shorter OS in HR+/HER2- tumors (HR: 1.10; 95% CI: 1.02–1.19; *p* = 0.011) [15,38]. The meta-analysis conducted by Gao et al. confirmed this association (HR: 1.08; 95% CI: 1.02–1.14; *p* = 0.012). However, in both cases, TILs were not prognostic for disease-free survival (DFS).

As far as the predictive role of TILs for NACT in terms of pCR, this is also controversial. In fact, the pooled analysis of the German Breast Cancer Group demonstrated a statistically significant association between TILs and pCR, which was achieved in 6% of patients with low TILs (0–10%), 11% of patients with intermediate TILs (11–59%), and 28% of patients with high TILs (60–100%) (*p* < 0.0001, χ^2^ test for trend). Gao et al., instead, did not confirm this association in their meta-analysis (OR: 1.15; 95% CI: 0.79–1.69; *p* = 0.460) [15,38].

The signals of an inverse prognostic role of TILs emerging in this subset of BC may be further investigated considering the high heterogeneity of HR+/HER2+ BCs, which includes two different intrinsic subtypes, namely luminal A and luminal B tumors, respectively characterized by a higher and lower dependence on endocrine signaling and by different prognoses [53,54]. Therefore, the comprehension of luminal immune TME requires a deeper dissection of both the TIL landscape and the tumor intrinsic subtype. In fact, it has been shown in the early setting that the prevalence of TIL positivity (defined as TILs ≥ 5%) is higher in luminal B-like subtypes than in luminal A-like ones (28% vs. 16%; *p* > 0.0001) [48].

When focusing on the TILs subpopulations, the association between a higher CD8+ TILs infiltration and a reduction in BC-specific survival is conserved [49,55]. Instead, considering FOXP3+ cells, which include both CD4+/FOXP3+ cells (Treg) and CD8+/FOXP3+ cells (activated CD8 T cells), the evidence is discordant, maybe because this biomarker is expressed by immune cells involved both in tumor suppression and in tumor progression [56,57]. In fact, high levels of FOXP3+ TILs are associated with poor survival in HR+/HER2- BCs that lack CD8+ TILs [14].

Notably, a recent retrospective analysis including 563 patients with early HR+/HER2- BC showed that CD8+ sTILs were higher in patients with PIK3CA-mutated tumors (OR: 1.65; 95% CI: 1.03–2.68) and that, in this PIK3CA-mutated subpopulation, they are associated with a higher risk of recurrence (HR: 1.98; 95% CI: 1.14–3.41) on multivariate analysis, while CD4+ and FOXP3+ sTILs failed to demonstrate a prognostic value [52]. This association further highlights the importance of considering both intrinsic and extrinsic features to better understand the immune landscape of this BC subtype.

Lastly, the most intriguing data derive from the correlations between tumoral features and the immune landscape. In fact, TILs and immune-related genes in HR+/HER2- BC have been associated with a poor response to therapy with aromatase inhibitors as well as lower *ESR1* expression levels, which are shown to be mutually exclusive with *TP53* mutations in metastatic endocrine-resistant HR+/HER2- BC [58,59,60,61,62]. Furthermore, *TP53* mutations are known to be more prevalent in HR+/HER2- BC with the PAM50-basal intrinsic subtype and are associated with a higher infiltration of CD8+ T cells and B cells [62,63]. Considering these data, the higher prevalence of TILs in the luminal B-like subtype, and the prognostic value of TILs in luminal-like BCs, TILs may be a marker of luminal-like BCs with an intrinsic basal subtype.

In conclusion, there is no role for routine assessment of TILs in early HR+/HER2- BC, and their presence cannot be used to guide prognosis or as a predictive biomarker [7]. A better dissection of luminal immune TME, distinguishing immunosuppressive from immune-activating cells, and of tumor features, evaluating intrinsic subtypes, may improve the comprehension of this scenario.

### 3.4. Clinical Validity and Levels of Evidence

Most of the data presented deal with the assessment of the prognostic value of TILs in patients with early-stage BC and can be categorized into different levels of evidence.

According to levels of evidence for biomarkers redefined by Simon and colleagues [64], level 1 evidence for a biomarker requires the demonstration of its validity in an appropriately powered prospective study that is specifically designed to test the biomarker or in a meta-analysis and/or overview of level 2 or 3 studies; instead, level 2 evidence requires a study in which marker data are determined in a prospective trial not specifically designed to test marker utility [65]. The analysis can be performed on samples prospectively collected in a clinical trial designed to address tumor markers (A), on archival material from independent randomized trials (B), on archival material prospectively collected in an observational register (C), or on specimens collected for other reasons, processed, and stored for secondary analyses (D) [66].

Using these widely accepted criteria and taking into account the consistency of results from available pooled- and meta-analyses, the clinical validity of TILs as a prognostic biomarker in early-stage BC can be established at 1B for TNBC and 2B for other subtypes (Table 1).

Next steps in clinical validity assessment require the definition of a threshold for each BC subtype, a better characterization of TILs’ prognostic role in early HR+/HER2- BC, the definition of the prognostic value of TILs in advanced disease, and, lastly, the assessment of the predictive role of response to IO agents across all subtypes and all stages. All these issues require prospective validation in cohorts stratified according to TILs’ value.

## 4. Assessment of TILs Clinical Utility

The clinical utility of TILs assessment, that is, TILs’ ability to guide treatment decisions and improve outcomes, has not been clearly established.

The assessment of the clinical utility of TILs requires clinical trials with TILs as a biomarker for patient selection or patient stratification. In the case of patient selection, only patients with TILs-positive or TILs-negative disease are randomized, but analytic and clinical validity must have been robustly assessed before; in the case of patient stratification, patients with both TILs-negative and TILs-positive disease are randomized; such an approach requires four-arm randomized clinical trials with higher sample sizes, but would more strongly provide evidence of clinical utility, especially for biomarkers of uncertain significance [67,68].

The most important questions about the clinical utility of TILs comprise the selection of responders to IO agents across all BC subtypes and the possibility of chemotherapy de-escalation, especially in patients with early TNBC, in order to reduce toxicity. According to these clinical questions, appropriate prospective studies supporting the clinical validity of TILs in the setting of clinical utility are required. Pending these data, guidelines and the International TILs Working Group do not currently recommend the routinary use of sTILs to guide treatment decisions in clinical practice (e.g., type and duration of cytotoxic chemotherapy) [7].

Current trials assessing the clinical utility of TILs in BC are reported in Table 2. Only one of them (NCT05556200) is evaluating a de-escalation approach in patients with stage II-III TNBC and TILs > 10%, which consists of a NACT-free regimen (camrelizumab plus apatinib) with physician’s choice chemotherapy allowed only to patients not achieving the pCR.

NCT05491226 and BELLINI are two non-randomized window of opportunity studies; the first one is evaluating an escalation approach with pembrolizumab, radiation therapy, and a CSF-1R inhibitor, prior to standard treatment, in patients with node-positive or less immunogenic (TILs ≤ 40% or PD-L1-negative) non-metastatic TNBCs. BELLINI trial, the first clinical trial selecting patients according to TILs, to orient an escalation approach with nivolumab and ipilimumab. Results from a cohort of the trial have been presented at ESMO Congress 2022: 30 patients with stage I-III TNBC and TILs ≥ 5% were included [69]. It has been shown that a window opportunity with 2 cycles of nivolumab with or without ipilimumab is respectively associated with an ORR of 19% and 27%, and with a 2-fold increase in CD8 T cells and/or IFN-γ expression in 53% and 60% of patients.

The three clinical trials in the metastatic setting (BELLA, PERICLES, and MIMOSA) are selecting or stratifying patients according to TIL status in order to enrich potential responders to ICIs (Table 2).

## 5. Future Challenges and Conclusions

Current issues and future challenges related to the assessment of TILs analytical validity, clinical validity and clinical utility are summarized in Figure 1.

The most important challenges comprise: the further characterization of the BC immune TME, also through more sophisticated approaches, such as analyzing, for example, the functional and maturation states and metabolism of TILs [70]; the elucidation of the interactions between tumor cells and immune cells across all subtypes.

Furthermore, the analytical validity may be implemented through web-based platforms aiming at enhancing the interaction among pathologist and increasing the concordance of morphological assessment, as demonstrated in TONIC trial (NCT02499367), in which concordance values between four pathologists were >90%; and through the development of operator-independent approaches (e.g., machine-learning-based) [71,72,73].

Apart from the issues discussed above, the assessment of clinical validity may be further enriched by exploring novel time points and novel ways for TILs evaluation. For example, a meta-analysis of 22 studies, including 1569 patients with early BC who received TIL dynamic assessment during NACT, demonstrated that an increase in TILs in TNBC was associated with better DFS in univariate analysis (HR: 0.59; 95% CI: 0.37–0.95; *p* = 0.03) [74]. Furthermore, TILs levels were assessed in patients with TNBC who had residual disease (RD) after NACT, and an association with improved RFS and OS was discovered [75]. However, the combination of TILs in RD with a residual cancer burden (RCB) class was shown to predict BC recurrence across all subtypes [76].

In conclusion, TILs represent a promising and inexpensive prognostic and predictive biomarker to be prospectively validated in clinical trials, which may help physicians to select patients with BC who may better respond to IO agents and/or deserve chemotherapy de-escalation. Therefore, the use of TILs for guiding treatment decisions is not supported at the moment for standard clinical practice, but it is paramount in the context of TILs-informed randomized clinical trials.

## Figures and Tables

**Figure 1 cancers-15-00767-f001:**
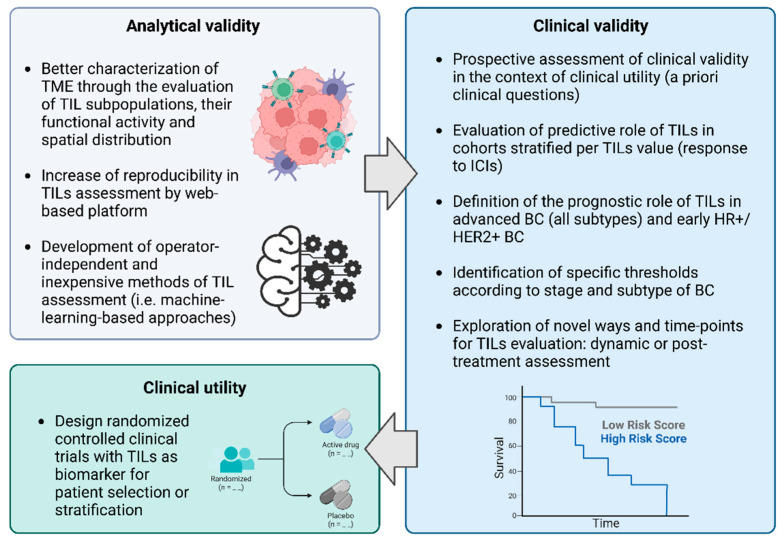
Current issues for the assessment of the analytical validity, clinical validity, and clinical utility of TILs in BC. Keys: BC, breast cancer; HER2, human epidermal growth factor receptor 2; HR, hormone receptor; ICIs, immune checkpoint inhibitors; TILs, tumor-infiltrating lymphocytes; TME, tumor microenvironment. Created with BioRender.com (accessed on 20 October 2022).

**Table 1 cancers-15-00767-t001:** TILs prognostic value with level of evidence across BC subtypes (early setting).

Endpoint	TNBC	HER2+ BC	HR+/HER2- BC
pCR	Positive correlation (1B)	Positive correlation (2B)	Discordant data
EFS/DFS	Positive correlation (1B)	Positive correlation (2B)	No correlation
OS	Positive correlation (1B)	Positive correlation (2B)	Negative correlation (2B)

Key: BC, breast cancer; DFS, disease-free survival; EFS, event-free survival; HR, hormone receptor; OS, overall survival; pCR, pathological complete response; TNBC, triple-negative breast cancer.

**Table 2 cancers-15-00767-t002:** Clinical trials assessing the clinical utility of TILs in breast cancer.

Trial and NCT Number	Ph.	Patients	Design	Endpoints
**Early Breast Cancer**				
NCT05556200	2	Stage II-III TNBC with TILs > 10%	Neoadjuvant Camrelizumab (anti-PD-1) + Apatinib (VEGFR2 inhibitor)	pCR
NCT05491226	2	cM0 TNBC with TILs ≤ 40% or cN+ or PD-L1-	Pembrolizumab + RT + Axatilimab (CSF-1R inhibitor), followed by SOC curative-intent treatment (NACT or surgery)	pCR
BELLINI (NCT03815890)	2	Stage I-III TNBC with TILs ≥ 5%	Nivolumab with or without Ipilimumab for 2 cycles followed by SOC curative-intent treatment (NACT or surgery)	Immune- activation
		cN0 early TNBC with TILs ≥ 50%		
		Stage I-III luminal B-like BC with TILs ≥ 1%		
**Advanced or Metastatic Breast Cancer**				
BELLA (NCT04739670)	2	mTNBC with PD-L1+ or TILs ≥ 5%, DFI < 12 mo	First line carboplatin + gemcitabine + bevacizumab + atezolizumab until PD	PFS
PERICLES (NCT03971045)	2	Lymphangitic pretreated and inoperable BC with PD-L1+ and/or TILs ≥ 1%	Pembrolizumab + metronomic cyclophosphamide until PD	ORR
MIMOSA (NCT04307329)	2	Pretreated HER2+ mBC with low (<5%) or high (≥5%) TILs	Monalizumab + Trastuzumab until PD, in patients with high TILs (cohort A) and low TILs (cohort B)	ORR

**Keys**: cN, clinical nodal status; CSF-1R, receptor of the colony-stimulating factor-1; DFI, disease-free interval; HER2, human epidermal growth factor receptor 2; mBC, metastatic breast cancer; mo, months; NACT, neoadjuvant chemotherapy; ORR, objective response rate; pCR, pathological complete response; PD, progression of disease; PD-(L)1, programmed death ligand 1; PFS, progression-free survival; Ph, phase; RT, radiation therapy; SOC, standard of care; TILs, tumor-infiltrating lymphocytes; (m)TNBC, (metastatic) triple-negative breast cancer; VEGFR2, vascular endothelial growth factor receptor 2.

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
