# Peer review of "Tumor Infiltrating Lymphocytes across Breast Cancer Subtypes: Current Issues for Biomarker Assessment"

_cancers, 2023, doi:10.3390/cancers15030767_

Round 1

Reviewer 1 Report

Thank you for the chance you gave me to read this interesting review paper entitled “Tumor Infiltrating Lymphocytes across Breast Cancer subtypes: current issues for biomarker assessment” by Valenza et al.  In this study, the authors give a thorough overview of the clinical value of tumor infiltrating lymphocytes (TILs) in breast cancer as well as they present the most important current issues and future challenges regarding their potential integration into the clinical practice. This clinically oriented topic has great importance, the manuscript is well-written and structured, and the text has a good flow with the exceptions of “simple summary” and “abstract”, which could be improved. A significant issue with this manuscript has to do with the high score (37%) based on the plagiarism detection service “Turnitin”, which needs to be treated before being suitable for publication.

Author Response

Thank you for your suggestion. We emended the manuscript according to your suggestions and rephrased many highlighted sentences at risk of plagiarism.

For example: “In fact, the improvement of outcomes through the optimization of therapeutic strategy requires validated biomarkers for patient selection and to guide treatment (de)escalation. The introduction of a biomarker requires the demonstration and assessment of: (i) the assays’ technical performance, namely analytical validity; (ii) the ability of the biomarker to identify relevant clinical subgroups, with specific prognostic and/or predictive features, namely clinical validity; and (iii) the ability of the biomarker to guide treatment decisions providing a clinical benefit, namely clinical utility.[6]” (pag 2)

Or: “The assessment of clinical utility of TILs requires clinical trials with TILs as a biomarker for patient selection or patient stratification. In the case of patient selection, only patients with TILs-positive or TILs-negative disease are randomized, but analytic and clinical validity must have been robustly assessed before; in case of patient stratification, patients with both TILs-negative and TILs-positive disease are randomized: such approach requires a four-arm randomized clinical trials with higher sample sizes, but would more strongly provide evidence of clinical utility, especially for biomarkers of uncertain significance” (pag 8)

Or: “prospective studies supporting the clinical validity of TILs in the setting of clinical utility are required. Pending these data, guidelines and the International TILs Working Group do not currently recommend the routinary use of sTILs to guide treatment decisions in clinical practice” (pag 8)

Reviewer 2 Report

  1. This is a review paper presenting many studies of tumor-infiltrating lymphocytes in breast cancer. I think "literature review" should be in the title.
  2. I thought the overall organization of the paper was fine, but I was overwhelmed by the facts and figures. Perhaps moving the content to tables or figures would help the reader compare common characteristics (study size, thresholds, drugs, results). I believe I was asked to review this paper to provide an assessment of methods and statistics. However, there were none employed. This paper would benefit from an attempt to harmonize results across studies, specifically the metric of performance and the threshold used.
  3. The writing would benefit from a review of grammar and English. The sentences were overly complicated and long with many commas, making them hard to digest. I would recommend shorter sentences. There were problems with prepositions (to, at, with …), pronouns (who, what, which, …), and awkward verb use. Here’s some examples:
    1. On page 1: “Immunotherapy (IO) drugs and in particular immune-checkpoint inhibitors (ICIs) have revolutionized the treatment landscape of several cancer types, including TNBC, despite BC has historically been considered immunogenically quiescent, hence less likely to derive benefit from IO approaches [4].” The phrase “despite BC has historically been considered” is awkward verb usage, probably because of the long sentence. I suggest splitting the sentence or changing the awkward part to, “despite BC having been historically considered”.
    2. On page 2: “Tumor infiltrating lymphocytes (TILs) represent a surrogate biomarker of anti-tumor, lymphocyte-mediated immunity, which activation is related to better outcomes in patients with TNBC [5]. Enrichment in TILs may identify highly immunogenic BCs and orient toward immuno-oncology treatments, while identifying patients at particularly better prognosis, who may need de-intensified treatments, for example less chemotherapy or no chemotherapy.” I don’t think “which” is being used properly in the phrase, “which activation is related to better outcomes”.  I don’t know what the direct object is for the verb “orient”: who is being oriented? The preposition “at” following patients should be “who have better prognosis” or “with better prognosis”.
    3. On page 4: “Despite these data may suggest …” I do not understand the role of “despite” here. Would the sentence be ok without “despite”?
    4. On page 5: “… in which anyway sTILs were not specifically related …” I do not understand the role of “anyway”. Can you delete it?
    5. On page 5 you are talking about prospective validation. How come “ad hoc” appears in this sentence. That seems wrong.
  4. On page 3 you say “nonetheless, the term lymphocyte-predominant BC (LPBC) is usually used to describe BC with more than 50% of stro-mal lymphocytic infiltration.” This deserves a reference.
  5. On page 3 you say, “concordance … is very high”. This language is overly subjective. I ask that you provide a metric and its value.
  6. On page 4 you say, “consensus from 2021 … do not endorse TILs as routine pathological marker.” I understand the opposite. Please check.

Author Response

This is a review paper presenting many studies of tumor-infiltrating lymphocytes in breast cancer. I think "literature review" should be in the title.

Thank you for your suggestion. We preferred not to introduce the term “literature review” in the title since in MPDI manuscripts it is already written above the title and because of the length of the title.

I thought the overall organization of the paper was fine, but I was overwhelmed by the facts and figures. Perhaps moving the content to tables or figures would help the reader compare common characteristics (study size, thresholds, drugs, results). I believe I was asked to review this paper to provide an assessment of methods and statistics. However, there were none employed. This paper would benefit from an attempt to harmonize results across studies, specifically the metric of performance and the threshold used.

Thank you for your suggestion. We preferred not to include a table because it is impossible to summarize the high heterogeneity of populations included in studies and their characteristics in a synoptic table. As far as the harmonization of analytical validity across studies, as stated, International TILs Working Group were used for TILs assessment in all cited studies.

The writing would benefit from a review of grammar and English. The sentences were overly complicated and long with many commas, making them hard to digest. I would recommend shorter sentences. There were problems with prepositions (to, at, with …), pronouns (who, what, which, …), and awkward verb use. Here’s some examples: On page 1: “Immunotherapy (IO) drugs and in particular immune-checkpoint inhibitors (ICIs) have revolutionized the treatment landscape of several cancer types, including TNBC, despite BC has historically been considered immunogenically quiescent, hence less likely to derive benefit from IO approaches [4].” The phrase “despite BC has historically been considered” is awkward verb usage, probably because of the long sentence. I suggest splitting the sentence or changing the awkward part to, “despite BC having been historically considered”.

Thank you for your suggestion. We emended the sentence according to your advice: “despite BC having been historically considered”

On page 2: “Tumor infiltrating lymphocytes (TILs) represent a surrogate biomarker of anti-tumor, lymphocyte-mediated immunity, which activation is related to better outcomes in patients with TNBC [5]. Enrichment in TILs may identify highly immunogenic BCs and orient toward immuno-oncology treatments, while identifying patients at particularly better prognosis, who may need de-intensified treatments, for example less chemotherapy or no chemotherapy.” I don’t think “which” is being used properly in the phrase, “which activation is related to better outcomes”.  I don’t know what the direct object is for the verb “orient”: who is being oriented? The preposition “at” following patients should be “who have better prognosis” or “with better prognosis”.

Thank you for your suggestion. We emended the sentence according to your advice: “Enrichment in TILs may identify highly immunogenic and immune-vulnerable BCs, as well as patients with better prognosis, who may need de-intensified treatments, for example less chemotherapy or no chemotherapy”

On page 4: “Despite these data may suggest …” I do not understand the role of “despite” here. Would the sentence be ok without “despite”?

Thank you for your suggestion. We emended the sentence according to your advice: “These data may suggest a predictive value of TILs in the response to ICI, however their interpretation requires caution.”

On page 5: “… in which anyway sTILs were not specifically related …” I do not understand the role of “anyway”. Can you delete it?

Thank you for your suggestion. We emended the sentence according to your advice: “In fact, most of these immune biomarker analyses were exploratory and TILs were not included among stratification factors, apart from GeparNuevo trial, in which sTILs were not specifically related to durvalumab-response.”

On page 5 you are talking about prospective validation. How come “ad hoc” appears in this sentence. That seems wrong.

Thank you for your suggestion. We emended the sentence according to your advice: “Furthermore, the prognostic and predictive role of TILs requires an independent prospective validation in randomized clinical trials.”

On page 3 you say “nonetheless, the term lymphocyte-predominant BC (LPBC) is usually used to describe BC with more than 50% of stro-mal lymphocytic infiltration.” This deserves a reference.

Thank you for your suggestion. We added the reference, which is the same of the previous sentence.

On page 3 you say, “concordance … is very high”. This language is overly subjective. I ask that you provide a metric and its value.

Thank you for your suggestion. We emended the sentence according to your advice: “The inter-pathologist concordance of the visual and inexpensive method proposed by the International TILs Working Group is very high (up to 86%)”

On page 4 you say, “consensus from 2021 … do not endorse TILs as routine pathological marker.” I understand the opposite. Please check.

Thank you for your suggestion. The sentence is correct. In fact, according to San Gallen International Breast Cancer consensus “However, the Panel again declined to endorse either of these approaches as routine pathological markers in early- stage TNBC. TILs appear to serve as a prognostic marker for response to neoadjuvant chemotherapy, but data are not considered adequate for choosing specific regimens or deciding whether to withhold chemotherapy treatment. PD-1/PD-L1 expression predicts benefit from addition of checkpoint inhibitors to chemotherapy in the treatment of metastatic TNBC. However, trials have not shown that PD-L1 expression predicts the improvement in pathological complete response (pCR) when checkpoint inhibitors are added to neoadjuvant chemotherapy, an approach which (as of this date) remains investigational for early-stage TNBC” (DOI: https://doi.org/10.1016/j.annonc.2021.06.023).

Round 2

Reviewer 1 Report

Although pdf file doesn't permit a better evaluation of similarity by using plagiarism detesction services (since it includes deleted sentences), it seems that the authors have improved this point. However, two first paragraphs of the introduction needs further improvement before publication.